# Targeted Large-Scale Genome Mining and Candidate Prioritization for Natural Product Discovery

**DOI:** 10.3390/md20060398

**Published:** 2022-06-16

**Authors:** Jessie James Limlingan Malit, Hiu Yu Cherie Leung, Pei-Yuan Qian

**Affiliations:** 1Southern Marine Science and Engineering Guangdong Laboratory (Guangzhou), Guangzhou 511458, China; jjmalit@connect.ust.hk (J.J.L.M.); hyleungaq@connect.ust.hk (H.Y.C.L.); 2Department of Ocean Science and Hong Kong Branch of the Southern Marine Science and Engineering Guangdong Laboratory, The Hong Kong University of Science and Technology, Hong Kong, China

**Keywords:** genome mining, secondary metabolites, natural products, bioactive compounds, antibiotics, genomics

## Abstract

Large-scale genome-mining analyses have identified an enormous number of cryptic biosynthetic gene clusters (BGCs) as a great source of novel bioactive natural products. Given the sheer number of natural product (NP) candidates, effective strategies and computational methods are keys to choosing appropriate BGCs for further NP characterization and production. This review discusses genomics-based approaches for prioritizing candidate BGCs extracted from large-scale genomic data, by highlighting studies that have successfully produced compounds with high chemical novelty, novel biosynthesis pathway, and potent bioactivities. We group these studies based on their BGC-prioritization logics: detecting presence of resistance genes, use of phylogenomics analysis as a guide, and targeting for specific chemical structures. We also briefly comment on the different bioinformatics tools used in the field and examine practical considerations when employing a large-scale genome mining study.

## 1. Introduction

Natural products (NPs), which are produced by bacteria, plants, fungi, animals, and other living cells, have been an inspiration for pharmaceutics and other biological agents such as herbicides and insecticides. In theory, NPs are structurally optimized to interact with biological targets after several hundred millennia of evolution [1] and, therefore, have a higher chance of being bioavailable than traditional small molecule drugs [2]. NPs are often categorized into primary and secondary metabolites. Primary metabolites include nutrients and building blocks for cellular maintenance, whereas secondary metabolites endow their producers with selective advantages in ecological niches and help them survive under dynamic environments [3]. For instance, some secondary metabolites function as antibiotics, pigments, and growth hormones [4]. Thus, secondary metabolite NPs have been targeted as novel drug leads with potent bioactivities, with more than 30% of approved small molecule drugs being bacterial secondary metabolites [5].

The synthesis of these compounds is catalyzed by enzymes. For bacterial and fungal secondary metabolites that had their genetic basis elucidated, the genes encoding such enzymes tended to cluster close together in the genome. These clusters are termed biosynthetic gene clusters (BGCs) [6]. Other relevant genes, such as those involved in the regulation, transport, and resistance to the self-produced compounds, can be found in or alongside the BGCs [7]. Biosynthesis enzymes of secondary metabolites have been extensively studied because of the unprecedented chemical transformations they can catalyze in different compound scaffolds [8]. Bacterial secondary metabolites can be classified according to their chemical scaffolds and signature biosynthesis enzymes that synthesize their core structures. Classes of secondary metabolites include polyketides, peptide-derived NPs, isoprenoids, terpenoids, alkaloids, nucleotide-based NPs, phenylpropanoids, and glycosylated NPs [9], which can be further divided into subclasses according to their additional features or chemical modifications.

Traditionally, the discovery of novel secondary metabolites with potent bioactivities from the pool of NPs produced by a bacterium or a microbial consortium involves a culture-first process. In this approach, the NP production occurs chronologically as follows: isolation of microorganism(s) of interest, culturing the microbe(s) under different conditions to induce NP production, and purification of desired products through chemical extraction. To test the presence of NPs with desirable bioactivities, bioassays are conducted on bacterial culture, extracted fractions, or purified products to study their effects on various organisms or human disease models, a process known as bioactivity-guided screening [10]. This traditional NP discovery pipeline does not require prior knowledge of a producer’s genome nor its biosynthesis capability, and, thus, it is termed the “top-down” approach [11]. Although the “top-down” approach has facilitated the discovery of numerous therapeutic NPs, it suffers from several pitfalls. For instance, it leads to the re-isolation or rediscovery of known compounds produced by different bacterial species, because multiple species can produce the same NP [12,13]. Moreover, some secondary metabolites are difficult to be detected, purified, and examined through bioassays because they are produced at trace concentrations. Cultivation under laboratory conditions may not provide all the environmental stimuli required to produce the desired secondary metabolites. A significant percentage of microflora have been nonculturable in the laboratory, making it difficult to identify their NPs [14]. 

An alternative to the “top-down” approach mentioned above is to search bacterial genomes for the NP biosynthesis machineries [15]. This genome-mining-based approach, known as the “bottom-up” approach, leverages the abundance and availability of genomic data, bioinformatics analysis, and genetic manipulation tools to search for BGCs encoding novel NPs [11]. These data-mining approaches often utilize signature genes and conserved protein domains of enzymes responsible for the biosynthesis of different types of NPs in their search. These enzymes have highly conserved amino acid sequences, despite their ability to generate diverse families of compounds; hence, they can be used as seeds or reference sequences for identifying gene homologs and, therefore, novel BGCs [16]. Before cheap and fast genome sequencing techniques became available, detecting such conserved sequences of target NPs in genomic libraries was achieved by using probes in Southern hybridization experiments or degenerate primers for PCR-based detection [17]. Today, gene homologs can be detected through in silico approaches [18], many of which utilize BLAST searches [19] or profile-based analyses such as HMMER searches to characterize the target NP BGCs and search for novel BGCs in genomic data [20]. The genomic approach also allows for the genetic manipulation of biosynthesis pathways to improve yield, activation of silent BGCs, and heterologous expression of BGCs in a host; it also provides a direct link between a metabolite and its genetic origins [21].

## 2. Genomics for Natural Product Discovery

Several software packages have been specifically developed to search for secondary metabolite BGCs in bacterial genomes. The most widely used and powerful software pipeline currently is the antibiotics and Secondary Metabolite Analysis Shell (antiSMASH), which uses manually curated gene cluster rules to identify core and additional biosynthesis genes with profile Hidden Markov Models (pHMMs) [22]. The latest version of antiSMASH supports the identification of more than 70 types of BGCs encoding for different chemical scaffolds [23]. The software can also predict the chemical structure of a core scaffold by using the substrate specificity predictions of protein domains found in polyketide synthase (PKS) and nonribosomal peptide synthase (NRPS) modules, and assumed collinearity [22]. ClusterBlast, one of the incorporated sequence alignment tools in antiSMASH, facilitates the comparison of BGCs; this comparison is necessary for the dereplication process, wherein previously known and characterized BGCs from the mining results are removed [24]. Other genome-mining tools developed for BGC detection include PRISM, which also uses pHMMs to detect BGCs and additionally predicts biological activity through machine learning [25]; and MetaBGC, which identifies BGCs from the sequence data of microbial communities [26]. The Secondary Metabolite Bioinformatics Portal (SMBP) provides a list of genome-mining software and includes links to software, tools, and databases relevant to NP discovery with omics data [27]. 

Technological advancements and the rapid increase in the amount of publicly available genome information in the previous decades have spurred many large-scale in silico genome-mining analyses. Instead of mining a single genome at a time, we can now incorporate big data approaches to mine a large pool of genomes or metagenomes. The large-scale, extensive search in whole-genome databases for novel NPs is an approach that we herein refer to as “global genome mining”. In addition to the manually curated BGC databases, and those from the National Center for Biotechnology Information (NCBI) [28], several publicly available online databases can facilitate the global genome mining analysis of BGCs. The antiSMASH database is a repository of antiSMASH-annotated BGCs from more than 20,000 bacterial genomes and contains more than 150,000 BGCs. The Integrated Microbial Genomes Atlas of Biosynthetic Gene Clusters (IMG-ABC) includes more than 400,000 BGCs detected in archaea, bacteria, fungi and metagenome bins [29]. The Minimum Information about a Biosynthetic Gene Cluster (MIBiG) database aims to collect all characterized BGCs with known functions and the secondary metabolites they produce [6]. This database is especially useful in identifying BGCs in sequenced genomes that can produce the same or similar sets of compounds according to sequence homology, thus preventing the rediscovery of known compounds. The MIBiG currently contains the BGC information of 1923 secondary metabolites. A more comprehensive list of databases that can aid large-scale BGC studies can be found in a recent review [30]. Handling these huge datasets requires the development of downstream bioinformatics platforms to properly display the uncharted NP diversity hidden in cryptic BGCs of genomes (Figure 1). Examples of such bioinformatics tools that can handle large-scale biological sequence data include EFI-EST [31], FastTree [32], and cd-hit [33]; they also have the ability to be integrated into the global genome mining pipeline for data analysis and visualization.

Several omics-based strategies have been developed for novel secondary metabolite discovery, including transcriptomics, proteomics, metabolomics, and meta-omics [16,34,35], as well as machine learning tools [36,37,38]. Their integration into the NP discovery pipeline has been discussed in previous reviews [16,34,35,36,37,38]. Given the immense, dark chemical space hidden in these cryptic BGCs and the large volume of hits that can result from a single round of global genome mining, one of the most significant challenges now is the prioritization of candidate BGCs that are worthy of further study. In the present review, we focus on natural product discovery studies that conduct large-scale targeted genome-mining approaches on manually curated or publicly available genome datasets. We present studies that have successfully uncovered novel NPs by using NP resistance determinants, phylogenomic analysis, or enzyme-catalyzed structural modifications to prioritize BGCs (Table 1). The aim of this review is to provide ideas and inspirations for researchers conducting large-scale natural-product genome-mining studies on how to identify candidate BGCs for further study.

## 3. Resistance-Gene-Guided Genome Mining

One approach in performing global genome mining for novel bioactive compounds is targeting biosynthesis-associated resistance determinants. Although most BGCs only contain genes responsible for synthesizing the desired secondary metabolite, some BGCs include genes involved in protecting the producing organism from the metabolite’s harmful effects [7]. These genes, called resistance genes, encode enzymes that impart immunity or resistance to the producer and generally work in three different ways. In product detoxification, the protein binds to and, in many cases, also modifies the produced secondary metabolite, thereby preventing the metabolite from binding to its target in a producer strain. An enzyme can also rapidly remove an NP from a cellular location or by binding it to a transporter. Lastly, a BGC can contain a duplicate resistant version of the NP’s target, or an enzyme that catalyzes the modification of the target, to render it resistant to a produced NP [7]. Several bioactive compounds and their modes of action have been identified by using resistance-guided genome mining (Figure 2). 

As mentioned, resistance genes can be colocalized with the BGC of the NP it protects it from. Griselimycin [67], salinosporamide A [68], and platensimycin [69] are examples of NPs discovered to contain resistance determinants in their BGCs. In genome mining, we can use previously confirmed biosynthesis-associated resistance genes as molecular “beacons”, illuminating BGCs that could produce novel compounds with similar bioactivities as the known NP. For instance, to search for novel topoisomerase inhibitors with anticancer and antibiotic qualities, Panter et al. [40] mined myxobacterial genomes for BGCs containing topoisomerase-targeting pentapeptide repeat protein (PRP) sequences. The choice of PRP was primarily inspired by cystobactamids, which contained PRP sequences in their BGC to protect the native gyrase from the cystobactamids. Their study led to the identification of pyxidicyclines–Type II polyketides with an unusual nitrogen-containing tetracene moiety. These compounds indeed inhibited bacterial topoisomerase IV, leading to its bactericidal effects, with an IC_50_ range of 6.25–1.6 μg/mL. The compounds had an MIC value of 0.06 μg/mL against HCT-116 cells, as well. The mode of action was believed to be through the inhibition of human topoisomerase I [40]. The same research group used PRP sequences again to prioritize Type III PKS candidate BGCs out of the 116 identified from myxobacterial genomes. A candidate BGC from *Cystobacterineae* strain MCy9487 was heterologously expressed in a myxobacterial model strain *Myxococcus xanthus* DK1622, which produced novel alkylpyrones that exhibited bacterial topoisomerase IV and human topoisomerase I inhibition, leading to cytotoxicity in HCT-116 cells but not bactericidal activity [39].

The enzymes responsible for transport of the produced compound can also be used to select for BGCs of interest. In a study focusing on the biosynthesis pathways of lipopeptides (LPs) produced by *Pseudomonas*, Girard et al. [42] identified a transport system dedicated to LP export by exploring the synteny of the LP-encoding genomes. This tripartite efflux system removes the antimicrobial LP from the *Pseudomonas* and is composed of three proteins, PleABC. These three genes are found near the LP-encoding BGC. The protein encoding for the inner membrane ABC transporter (PleB) was used as bait in BLASTP and antiSMASH analyses, and they obtained 75 putative LP-encoding BGCs from *Pseudomonas* genomes. Through a phylogenetic analysis of *pleB*, they identified four LP-encoding BGCs that contain *pleB* genes of various evolutionary distances from *pleB* in known LP BGCs. The strains containing these BGCs were subjected to crude extraction and chemical characterization, which led to discovery of a novel LP, prosekin. 

This resistance-gene-guided genome-mining approach has also been applied to genome mine fungal genomes. Dihydroxyacid dehydratase (DHAD), an enzyme involved in the branched chain amino acid biosynthesis pathway in plants, is essential to plant growth [70]. The absence of an equivalent pathway in animals suggests that this enzyme can be used as a herbicide target. To identify NPs with DHAD-inhibiting activities, Yan et al. [41] hypothesized that DHAD inhibitor genes are likely to appear alongside an inhibitor-resistant DHAD. They scanned fungal genomes for core biosynthesis enzymes that co-localized with DHAD and identified a cluster containing a sesquiterpene cyclase, two cytochrome P450s, and a DHAD homolog. This BCG was heterologously expressed in *Saccharomyces cerevisiae* RC01, producing aspterric acid. Extensive bioactivity testing showed that aspterric acid inhibited DHAD. Spray treatment on wild-type *Arabidopsis thaliana* drastically impeded plant growth and pollen formation. Transgenic *A. thaliana* containing a copy of this cluster’s DHAD gene was unaffected by the aspterric acid treatment, showing that it was indeed an inhibitor-resistant DHAD [41]. Using the same self-resistance gene logic, their group used a cytochrome P450 protein — lanosterol 14α-demethylase (CYP51), as a query in another search for bioactive compounds. CYP51 is involved in the ergosterol biosynthesis pathway and could be an effective target for anticholesterol and antifungal drugs. The group developed an algorithm to identify BGCs that have their core biosynthesis enzymes colocalized with CYP51, and identified a candidate. After initial confirmation of CYP51 activity of the candidate BGC through in vitro assays, heterologous expression was performed, uncovering the biosynthesis pathway of restrictin and lanomycin [43].

In a study to identify novel resistance determinants found in BGCs, our laboratory identified D-stereospecific resistance peptidases that act on D-amino acid–containing nonribosomal peptides (DNRPs). In the global genome mining pipeline that we developed, a networking analysis of biosynthesis elements, transporters, and regulators was applied to 2511 DNRP-encoding BGCs to identify novel resistance determinants. We found that D-stereospecific peptidases were strongly associated with DNRP biosynthesis genes and selected their coincident BGCs for further evaluation. Two candidate strains, *Brevibacillus laterosporus* DSM 25 and *Paenibacillus polymyxa* CICC10580, produced the DNRPs bogorol and tridecaptin, respectively. In the biochemical characterization of the associated D-stereospecific peptidases in each BGC, we showed that the two enzymes can cleave their respective DNRPs [45]. 

A phenomenon that is suggested to confer self-resistance is gene duplication, as it increases the number of the antibiotic target, thereby freeing up evolutionary pressure to allow the target to mutate into a resistant form. Genes that have undergone duplication can be identified by constructing a core genome that is conserved throughout a bacterial taxon and searching the individual pangenomes for additional gene copies, suggesting a gene duplication or horizontal gene transfer event. In a study by Tang et al. [44], they probed *Salinispora* genomes for duplicated housekeeping genes. They did this by identifying all the conserved orthologous groups (OGs) of protein-coding genes in *Salinispora*, and identified those that were shared among all strains to form the ‘core genome’. The OGs were grouped into clusters (COGs) to denote their general biological function. With a list of COGs and their hypothetical general functions, they then searched the pangenomes for additional OGs that allegedly had the same function as the OGs in the core genome, which could be duplicated OGs. Their pipeline correctly detected the presence of a duplicated 20*S* proteasome β-subunit present in the salinosporamide BGC, showing that their method was viable. Although duplication was detected in multiple COGs in their probing of *Salinispora*, they focused on a COG assigned to lipid transport and metabolism. The bacterial fatty acid synthase, in particular, was deemed favorable, as it displayed significant structural differences to its equivalent in mammals, implying low cytotoxicity of the produced compound. They discovered thiolactomycins and thiotetroamides by identifying a hybrid PKS-NRPS BGC that contained a duplicated fatty acid synthase gene. Indeed, the observed antibiotic activities of these families of compounds were owed to fatty acid synthase inhibition [44]. Something to be taken into consideration before using gene duplication phenomenon for NP target discovery, however, is that gene duplication events do not guarantee duplicated genes with new functions, because they may only serve metabolic redundancy and are not recruited to secondary metabolism pathways [71]. 

Although enzymes that were previously confirmed as resistance determinants provide a good starting point for a resistance-gene-guided genome-mining study, our study and that of Tang et al. showed that determining a specific resistant determinant beforehand is not necessary [44,45]. We can exploit the logic behind resistance mechanisms, instead of focusing on a known resistance gene, to discover new compounds and resistance mechanisms. Still, inspiration can be drawn from BGCs confirmed to contain resistance-related genes [7], or from compounds with an unknown biosynthesis pathway but with an established mode of action. Enzymes involved in their mode of action can be targeted and possibly be co-localized with a novel producing BGC. Resistance-gene databases, such as The Comprehensive Antibiotic Resistance Database [72] and Resfams [73], can be used to identify targets for genome mining. Prioritizing candidate BGCs associated with resistance determinants is also advantageous, as compounds identified with this pipeline are more likely to display bioactivities because their mode of action is hypothesized through the biological function of the resistance genes. 

## 4. Phylogenomics-Guided Genome Mining

As the resistant-target-based mining in the previous section already suggested, enzymes that are involved in secondary metabolite biosynthesis are basically paralogs of enzymes involved in primary metabolite generation [74]. In the same vein, these secondary metabolite biosynthesis enzymes can evolve and, thus, can lead to the catalysis of novel chemical transformations. This evolution generally occurs through the expansion of the enzyme’s substrate specificity [47]. As selective pressure directs these enzymes to produce metabolites with enhanced bioactivities, divergent enzymes can be used to identify BGCs encoding for NPs with novel chemical modifications and bioactivities [48]. This logic has been applied to discover novel NPs through the construction of phylogenetic trees to identify BGCs containing divergent biosynthesis enzymes, leading to the production of new compounds (Figure 3).

One of the first studies that used a global bacterial genome database for the analysis of secondary metabolism had used phylogenomics-guided mining and was undertaken by Fischbach’s laboratory. It developed the ClusterFinder algorithm to detect the BGCs of known and unknown NP classes. At the time of its creation, ClusterFinder raised interest when preexisting tools could only detect BGCs from characterized classes, but the software has since fallen out of favor for NP genome mining due to its limitations and the discovery that many of its hits were not valid secondary metabolite BGCs [23,75,76,77]. The algorithm uses an HMM-based probabilistic model trained on a curation of 732 BGCs that produce various known compounds. The genome sequence is converted into strings of PFAM domains by the software, and each domain is assessed for the probability that it belongs to a BGC, based on the training set used. Since the algorithm purely depends on PFAM domain frequencies and not on NP-specific genetic signatures, novel BGC classes can be detected. They applied this algorithm to 1154 bacterial genomes spanning the prokaryotic tree of life and performed a global phylogenomic analysis of all prokaryotic BGCs. They further constructed a BGC distance network by using evolutionary distances and identified groups of uncharacterized BGCs that are not detected by other genome-mining tools. They expressed representative BGCs from the group containing the greatest number of BGCs in *Escherichia coli*, producing aryl polyene carboxylic acids [46]. 

Cruz-Morales et al. [47] performed mining without targeting any particular NP class by identifying enzymes that have undergone enzyme expansion and repurposing from the central metabolism. These divergent enzymes are possibly recruited by secondary metabolite BGCs for the catalysis of chemical transformations devoted to NP biosynthesis. They started with identification of central metabolic enzymes and their orthologs from 230 actinobacterial genomes, and then they identified expanded enzyme families, selecting those with a higher number of orthologs than the average documented for enzyme families in the genome database. The identified expanded enzyme families were then searched against a database of NP-related enzymes from 226 known actinobacterial BGCs, and 23 expanded enzyme families putatively involved in NP biosynthesis were identified. Performing a phylogenetic analysis on each of the 23 enzyme families showed the presence of divergent clades, most of which contained homologs of enzymes from characterized BGCs. From these results, they focused on the 3-phosphoshikimate-1-carboxivinyl transferase family, AroA, and identified its associated BGC. The functional characterization of this BGC facilitated the production of a novel arseno-organic metabolite [47]. 

For known NP classes, researchers can target a specific gene or genes known to be conserved in known members of that class. Novel microviridins were isolated by using chemo-enzymatic synthesis by Ahmed et al. [49] after they prioritized their BGC hits with phylogenomics analysis. They identified 174 microviridin BGCs across the bacterial domain and then used the conserved MdnC ATP-grasp ligase gene to construct a maximum likelihood phylogenetic tree. The constructed tree revealed the presence of cryptic microviridin BGCs for further evaluation. They selected the microviridin BGC from *Cyanothece* sp. PCC 7822, as it putatively encodes for an unusual number of 10 precursor peptides. It also contains leucine, arginine, and lysine residues on critical positions that interact with serine proteases, thus suggesting that these microviridins could display serine protease inhibition. A combined chemo-enzymatic synthesis approach was used to successfully produce the new microviridins MdnA3, 6, 7, 8, and 9, with MdnA6 inhibiting trypsin at an IC50 of 21.5 μM, becoming the most potent microviridin-based trypsin inhibitor to date [49]. In a study by Culp et al. [48] on glycopeptide antibiotics (GPA), they constructed phylogenetic trees for not one conserved gene, but every common gene and gene segments from 71 GPA BGCs. The phylogenetic tree built from the condensation domains in the C2 module of nonribosomal peptide synthase—an enzyme involved in GPA production—revealed divergent clades from “true” GPAs. The “true” GPA BGCs possessed known GPA resistance genes, while those in the divergent clades lacked them. There were two such divergent clades, and a candidate was selected from one of the clades for BGC characterization. Subsequent fermentation led to the production of a novel compound corbomycin. Together with the characterized compound complestatin from the other clade, the two GPAs were shown to block autolysin activity required for cell wall development, a novel mode of action for GPAs, ultimately inhibiting bacterial growth [48]. 

A study by Yamada et al. on terpene synthases (TSs) found in bacteria was a case of conducting phylogenomics-guided mining on a gene that relatively lacks significant conserved sequences across species. As opposed to the conserved biosynthesis enzymes of other NPs, TSs originating from bacteria do not have significantly conserved domains and, thus, are not amenable for detection through sequence similarity such as in BLAST. A combined HMM and PFAM search method was thereby developed to mine for TSs found in bacterial genomes [78]. By mining from more than 8 million bacterial proteins, they were able to identify 262 proteins putatively encoding for TSs. They aligned and constructed a phylogenetic tree from these sequences to assign functions to the TSs found in several clades, such as geosmin and epi-isozizaene synthases. They were especially interested in isolated clades without a characterized member and prioritized them for heterologous expression in a *Streptomyces avermitilis* host. This has led them to isolate 13 new compounds—two sesquiterpenes, which were named hydropyrene, and its derivative, hydropyrenol; and 11 diterpenes [52]. 

A similar approach was used by Chen et al. in their study, wherein a global phylogenetic tree was constructed for the chain length factor (CLF) protein, a domain partly responsible for condensation reactions in Type II PKS synthases. They predicted the chemical class and the uniqueness of the polyketide structure that BGCs containing CLF proteins produced. To validate their results, several candidates that were evolutionarily distant from clades containing characterized proteins were selected for compound production. They characterized a novel polyketide oryzanaphthopyran, which contains an unusual angular naphthopyran scaffold [53].

Many NP global genome mining studies used phylogenetic mapping to examine their results. Most of the studies described in the present review have conducted phylogenetic mapping on candidate BGCs or genes. The exercise of constructing a phylogenetic tree for a quorum sensing protein led to a serendipitous NP BGC discovery, as reported by Mullins et al. [50]. Mullins et al. aimed to investigate the mechanisms behind the biocontrol activities of the noted pathogen killer *Burkholderia ambifaria* by analyzing the genomes of 64 *B. ambifaria* strains. They identified the BGCs in those genomes that putatively coded the known secondary metabolites of *B. ambifaria*. Of the BGCs, they constructed a phylogenetic tree for LuxR, a protein involved in quorum sensing system LuxRI, as it has been observed that the expression of BGCs can be regulated by quorum sensing. They discovered a candidate from the tree that featured a PKS BGC downstream of a LuxRI system. The characterization of this BGC led to production of cepacin A, a known potent anti-oomycetal that previously had unknown BGC origins. They further demonstrated that cepacin A inhibited the growth of the oomycete *Pythium ultimum*, which causes the damping-off disease in germinating crops [50]. 

Navarro-Muñoz et al. [51] developed two applications to aid in the analysis of huge BGC datasets: the Biosynthetic Gene Similarity Clustering and Prospecting Engine (BiG-SCAPE) and the Core Analysis of Syntenic Orthologues to Prioritize Natural Product Gene Clusters (CORASON). BiG-SCAPE groups BGCs identified by antiSMASH into gene cluster families (GCFs) by sequence similarity, whereas CORASON establishes phylogenetic relationships between the detected BGCs and generated GCFs. They used these two applications to mine 3080 actinobacterial genomes for detoxin and rimosamide BGCs. These BGCS were organized into GCFs by BiG-SCAPE, uncovering a conserved set of core genes, specifically *tauD*, which is present in all known detoxin/rimosamide-related BGCs. CORASON then constructed phylogenetic trees from the *tauD*-containing BGCs and revealed unexplored clades for further analysis. An LC–MS/MS metabolomics dataset that contained strains harboring BGCs from these unexplored clades was analyzed through molecular networking, resulting in the identification of 99 putatively novel detoxin or rimosamide analogs. They focused their characterization efforts on three detoxin BGC clades containing interesting characteristics for NP production: One of the clades has BGCs containing putative cytochrome P450 and enoyl-CoA hydratase/isomerase genes, leading to the production of detoxin S_1_, which contained an additional heptanamide side chain. The second clade had the detoxin BGCs adjacent to a spectomycin BGC, resulting in the identification of detoxins N_1_–N_3_, which feature a N-formylated tyrosine instead of phenylalanine. The third clade composed of *Amycolatopsis* BGCs containing a cytochrome P450 gene, revealing detoxins P_1_–P_3_, a diverse set of detoxins incorporating different amino acids. 

A phylogenomics-based mining approach incorporates different evolutionary theories that can supplement sequence-similarity and rule-based genome-mining approaches. It searches for divergent secondary metabolism enzymes or BGC features and, thus, can aid in identifying compounds with novel chemical scaffolds [46,47]. However, phylogenetic tree construction is not guaranteed to work for all enzymes or NP classes, and not all divergent enzymes catalyze new chemical reactions. In addition, the line between secondary metabolism enzymes and primary metabolism enzymes is not always clear, because of our limited understanding of gene evolution. Given this limitation, it is highly recommended to avoid the most common genes and to select slightly more uncommon genes for phylogenetic analysis [71]. Focusing on established core biosynthesis enzymes of NPs could help alleviate this drawback, as well. This is less likely to lead to the discovery of novel NP scaffolds, although there are examples of this happening.

## 5. Structure-Guided Genome Mining

In large-scale genome-mining studies, predictions on the structure of the NP core scaffold and predictions on its additional tailoring steps can show the possible chemical diversity of secondary metabolites produced by the uncharacterized BGCs. The predictions can also be used to discover novel enzymes that catalyze unprecedented chemical transformations. Thus, enzymes known to catalyze intriguing chemical reactions on NPs have been used to query and prioritize BGCs for interesting NP structures or novel NPs (Figure 4). 

Over 10,000 actinobacterial species were screened by Ju et al. [59] for the presence of phosphoenolpyruvate mutase gene *pepM*, which is an essential gene in the biosynthesis of phosphonic acids. Draft genomes were first obtained for the 278 strains containing *pepM*. The associated BGCs containing the gene were clustered together through a networking analysis, which uncovered uncharacterized phosphonic acid–producing GCFs. They selected four uncharacterized groups for further study and discovered three antibacterial phosphonopeptides, namely argolaphos AB and valinophos, along with the sulfur-containing phosphonates phosphonocystoximates and H-phosphinates.

Our group performed global genome mining of 162,672 bacterial genomes to search for novel cytochrome P450-associated cyclodipeptide (CDP) synthase BGCs. Cytochrome P450s have been previously shown to be able to catalyze a variety of chemical transformations on CDPs [79]. We found 829 BGCs that had such an association and selected one BGC from *Saccharopolyspora hirsuta* DSM 44795 that contained two cytochrome P450 genes, as we suspected that the two enzymes can perform sequential reactions on the CDP precursor. The heterologous expression of this BGC led to the description of the novel cyclodipeptides cyctetryptomycins A and B, which contain an unusual large macrocyclic core. Further bioassay testing showed that these compounds display potent neuroprotective activity [55]. Following this study, we were inspired by how prenylation can enhance the bioactivities of compounds and aimed to uncover new prenylated compounds by following the same pipeline. In this case, we targeted prenyltransferase-associated CDP synthase BGCs. There were substantially less of these BGCs than those containing cytochrome P450s, with only 26 BGCs detected. Surprisingly, a BGC from *Streptomyces griseocarneus* 132 that contained two prenyltransferase genes downstream of the CDP synthase gene was identified, akin to our previous CDP study. The BGC was then heterologously expressed, producing prenylated cyclodipeptides. We named this new family of compounds griseocazines, and a preliminary structure–activity relationship assay revealed that one of the griseocazines, the multiprenylated griseocazine D1, showed significantly improved neuroprotective activity compared to its nonprenylated counterpart [56].

Guangnanmycins and weishanmycins are leinamycin-type NPs discovered after large-scale genome mining by Pan et al. [58]. Leinamycin (LNM) contains a rare 1,3-dioxo-1,2-dithiolane moiety and has very potent antitumor activity, even in drug-resistant tumors. Its biosynthesis also involves a unique enzymology, involving a Type I PKS that lacks an acyltransferase domain; a bifunctional acyltransferase/decarboxylase responsible for β-alkylation; and an unusual domain of unknown function (DUF) and a cysteine lyase domain (SH), which catalyzes the incorporation of sulfur into the LNM backbone. In an effort to discover novel LNM analogues that also contained such sulfur incorporation, they identified *lnm*-type gene clusters by using the domains DUF–SH as a probe. They identified 19 putative *lnm*-type BGCs from 48,780 bacterial genomes and an additional 30 BGCs from their in-house culture collection of 5000 strains. All of the 49 identified strains were subjected to fermentation, and the two compounds were discovered.

In a sactipeptide study from Hudson et al. [57], the sequence similarity network of putatively identified rSAM maturases of sactipeptide and sactipeptide-like SCIFFs (six cysteines in forty-five residues [80]) and a maximum likelihood tree suggested that SCIFF-related rSAM maturases were closer to the QhpD protein family than they are to known sactipeptide rSAM maturases. QhpD enzymes catalyze S-Cβ and S-Cγ thioether linkages, but not the S-Cα ones that define the sactipeptide class. As it had been unclear previously whether SCIFFs possessed S-Cα linkages [81], this suggestion of SCIFFs possibly being not true sactipeptides inspired Hudson et al. to select a putative SCIFF, freyrasin, from *Paenibacillus polymyxa* ATCC 842 for characterization, and they were able to show that it did not have S-Cα links. They suggested the renaming of these sactipeptide-like peptides with non-S-Cα thioether bonds to radical non-alpha thioether peptides, or ranthipeptides [82]. This is another example of the benefit of large-scale mining allowing for serendipitous discoveries wherein unexpected patterns in the data occur and inform further investigations.

Our group used the predicted NP structure of candidates, directly selecting for NP features with theoretical bioactive functions to guide BGC prioritization of cationic nonribosomal peptides (CNRPs). Carrying positive charges, CNRPs can interact with the negatively charged bacterial cell membrane and have been demonstrated to have antibiotic activity against Gram-negative bacteria [83]. To analyze their diversity and discover novel CNRPs, we analyzed 7935 complete bacterial genomes for CNRP-encoding BGCs and identified 11,286 CNRP BGCs. We reduced the number of BGCs to 807, focusing on those with two or more positively charged residues in their CNRPs and discarding BGCs of siderophores and of shorter peptides. We then constructed a peptide-similarity network for these CNRPs, showing their diversity. Using this network as a guide, we identified clusters of uncharacterized putative N-acylated CNRPs—a group of lipopeptides, with members reported to be effective against Gram-negative bacterial infections. The presence of acyl groups on a hydrophobic moiety on N-acylated CNRPs can also impart membranolytic and cell-penetrating functions [84]. From the 261 putative N-acylated CNRPs, we decided to target those harbored in Bacilli because the taxon is a known producer for many CNRPs. We selected candidates with three or more positively charged amino acid residues, as they would theoretically show increased efficacy in disrupting the Gram-negative cell membrane. We finally produced three novel N-acylated cyclic depsipeptides: brevicidine, laterocidine, and paenibacterin B, which are broad spectrum Gram-negative antibiotics. We treated *E. coli* with the newly discovered CNRPs and observed the disruption of the outer cell membrane with atomic force microscopy [54].

As shown by these studies, a desired chemical moiety or enzymatic reaction can be used to target specific BGCs. Usually, a gene that produces an enzyme that catalyzes such chemical transformation is used as a bait, but conversely a predicted structure based on conserved protein domains could also be used, as in the case of NRPS and ribosomally produced peptides (RiPPs) (discussed in next section). The motivations for these structure-first prioritization approaches are usually the potent bioactivity imparted by the targeted chemical modifications, and to find analogs for compounds with a unique biosynthesis logic, or it could be simply to discover novel enzyme catalytic reactions. After filtering, phylogenetic and networking analyses can be used in conjunction with this approach in order to group BGCs with similar genetic architectures, thus exposing uncharacterized BGCs and allowing for the identification of divergent enzymes, both of which increase the likelihood of obtaining novel compounds.

## 6. Global Genome Mining for RiPPs

The mining of structure-modifying enzymes is especially useful for RiPP genome mining. RiPP BGCs consist of genes encoding precursor peptides and tailoring enzymes that catalyze post-translational modifications (PTMs) on the precursors, transforming them into mature peptide structures [82,85]. In contrast to nonribosomal peptides and polyketide synthetases, RiPPs have no universally conserved enzymes, enzyme domains, or other genetic features [64,86], but many recognized RiPP classes or families, such as lanthipeptide, sactipeptide, thiopeptide, and lasso peptide, have class-specific core enzymes and precursor peptide sequence motifs [82]. For genome-mining studies of RiPPs, the ribosomal origin of their precursor peptides facilitates the reliable prediction of their core structure solely on the basis of gene sequence [87,88]. Using the core tailoring enzymes of known RiPP classes to search for novel RiPPs is essentially the same approach as those described in the previous section on structure-guided genome mining. However, RiPP precursor peptides provide a challenge because their sequences are usually short and variable. RiPP precursors are therefore often unannotated in BLAST (see Reference [89], as an example). Hence, many genome-mining tools and software packages, including RODEO [89], RiPPER [64], and antiSMASH [22,23], have been designed specifically to identify these short precursor sequences. 

Many RiPP genome-mining approaches start with mining one or more core enzymes of an RiPP class, especially a class-specific tailoring enzyme, and then they search the surrounding genomic context for putative precursor sequences. An example of this approach to RiPP mining is the software RODEO, a tool that has since been incorporated into antiSMASH. RODEO characterizes the surrounding genomic context of a queried RiPP tailoring enzyme. It uses pHMMs to identify possible additional RiPP biosynthesis enzymes, followed by a precursor peptide search using RODEO’s ORF detection. The putative RiPP tailoring genes or precursors can then be clustered and analyzed, e.g., with a sequence similarity network. With this approach, the developers of RODEO have conducted searches for novel NPs from lanthipeptide [60], sactipeptide [57], thiopeptide [63], and lasso peptide [89] classes.

After clustering, most of these studies selected candidates from the resultant hits according to interesting or novel structures. They also considered the availability of the source strains. This resulted in the isolation of novel members of the respective RiPP classes (Figure 5). In their lasso peptide study, Tietz et al. [89] investigated candidates with novel predicted topologies, producing six novel lasso peptides and discovering novel and rare chemical modifications on lasso peptides—a new subclass featuring a novel topological position of a disulfide bond, and a lasso peptide that incorporated a citrulline. In their lanthipeptide study, Walker et al. [60] successfully produced a two-component lanthipeptide, birimositide, from *Strepotmyces rimosus* [90]. Hudson et al. [57] chose a candidate sactipeptide that had been previously bioinformatically predicted to have an interesting chemical structure and produced thuricin Z/huazacin, the fifth known sactipeptide [81,91].

As demonstrated by these studies, a working hypothesis is not required for candidate selection post-mining. Candidates can be selected simply on the basis of what appears rare, interesting, or unique, or vice versa, those which most resemble previously discovered compounds (see Reference [92], for example). In 2016, Skinnider et al. [62] mined for all known RiPP classes from 65,421 prokaryotic genomes and selected the rarest RiPP class from their clustering results, which had the least number of predicted members and only three documented members. This rare class was the YM-216391 family, with all three of the known members showing cytotoxicity activity at nanomolar levels and azole-rich macrocyclic structures. They isolated a new member of this rare class, named aurantizolcin. 

In their thiopeptide study, Schwalen et al. [63] further prioritized their putative thiopeptide hits post-RODEO. They targeted BGCs with a co-incidence of two enzymes—YcaO, which is a core enzyme of the thiopeptide class, and TfuA. TfuA–YcaO coincidence is required for a thioamidation PTM in *Methanosarcina acetivorans* [93], and was also found responsible for the thioamidated backbones of thioamitides [82,94,95,96]. Schwalen et al. examined a TfuA-like protein that appeared in 2% of their putative thiopeptide BGCs and selected a candidate from *Amycolatopsis*
*saalfeldensis* NRRL B-24474; they successfully extracted the novel thiopeptide saalfelduracin that possesses thioamidation PTM.

Santos-Aberturas et al. [64] directly used the aforementioned TfuA–YcaO coincidence to search for thioamidated RiPPs. They retrieved all the TfuA domain proteins from the actinobacterial phylum in the NCBI non-redundant protein sequence database, explored the genomic area around the *tfuA*-like gene for the *ycaO* gene, and used the genome mining tool RiPPer to search for precursor peptides. After constructing a peptide-similarity network from their hits, they focused on one orphan group and produced thioamidated peptides, named as thiovarsolins, from *Streptomyces varsoviensis*. 

Besides core enzymes and precursor sequences, other features in a BGC, such as operons, can be targeted. Bushin et al. [65] aimed to identify BGCs that featured streptide BGC-like rSAMs and were also under the control of *shp*/*rgg* quorum-sensing operons. Using a dataset of 2785 streptococcal genomes, they manually curated 592 rSAM-RiPP gene clusters and grouped the BGCs by the enzyme sequence similarity of rSAM. They functionally characterized members from the generated enzyme-similarity network and revealed unprecedented structural modifications possible on RiPPs, such as the formation of a tetrahydro [5,6] benzindole motif [97]; intramolecular β-thioether linkages [98]; aliphatic ether crosslinks [99]; arginine to tyrosine [100] and lysine to tryptophan [101] cross-links; and multiple sactionine macrocycles, such as in streptosactin [65].

Purushothaman et al. [61] appeared to skip the in silico precursor sequence mining process and instead used the tailoring enzyme of interest to narrow down their candidates. They were targeting a tailoring enzyme found in some cyanobactin BGCs, F-type prenyltransferases, and constructed a sequence similarity network from the entire InterPro family of around 100 cyanobactin prenyltransferases. Of these prenyltransferases, they selected a candidate TolF after constructing a Bayesian phylogenetic tree with known cyanobactin prenyltransferases. The Bayesian tree revealed that TolF was evolutionarily related to known cyanobactin prenyltransferase TruF1, but with only ~48% sequence similarity. The prospect of a functionally divergent TruF1-like prenyltransferase was intriguing because TruF1 catalyzed interesting reactions but was difficult to be characterized due to low solubility and low in vitro activity. They identified the other genes within the BGC of TolF and successfully produced novel cyanobactin tolypamide, in addition to a stable TolF prenyltransferase. Characterizing TolF revealed that it differed from TruF1 in the way it prenylates in the forward direction instead of reverse, thus granting researchers mechanistic insight into Ser- and Thr-prenylating enzymes when TruF1 was unavailable for extensive study [61].

RiPP BGCs tend to consist of a few genes. Some RiPP tailoring enzymes are substrate promiscuous, a phenomenon that some researchers have exploited. Van Heel et al. [66] selected lanthipeptide candidate BGCs that closely resembled those of the model lanthipeptide nisin. It was previously found that, when some lanthipeptide (or even non-lantipeptide) core peptides are fused to the nisin leader peptide, the hybrid lanthipeptide precursor peptide was accepted by the nisin biosynthesis system as a substrate, resulting in such hybrid BGCs producing non-nisin lanthipeptides [102]. Van Heel et al. wanted to exploit the promiscuity of nisin tailoring enzymes to engineer a convenient heterologous production system for producing novel lanthipeptides. They selected lanthipeptide candidate BGCs with only the same tailoring enzymes as nisin BGCs and successfully produced 31 peptides by using the heterologous production system. Moreover, five of the proteins showed antimicrobial activity, such as flavucin and agalacticin.

## 7. Conclusions

With the advancements in genome-mining technologies and development of the big data field, global genome mining for NP discovery has attracted considerable interest. The resultant large number of hits in a large-scale study often requires researchers to prioritize candidates for downstream investigations, because it is beyond the capabilities of many laboratories to screen every candidate. This review covers diverse approaches and tools for prioritizing candidates through global genome mining and discusses their advantages and limitations.

We introduced the studies that used resistance genes, phylogenomics, and NP chemical structures to guide the discovery of novel BGCs in large-scale genome mining. Resistance determinants are not found in all NP BGCs, but prioritizing candidate BGCs associated with resistance determinants is nevertheless a way to detect novel compounds. The approach of using a particular self-resistance enzyme or motif as molecular “beacons” has led to discoveries of NPs that putatively target the respective self-resistance enzymes [40,41]. Two different approaches that requires no prior knowledge of a resistance enzyme are to search for gene duplications and to search for cooccurrence of transport-related enzymes [42,44], as these phenomena suggest the possible presence of self-resistance mechanisms. The gene-duplication logic can be useful in identifying compounds with a specific bioactivity in mind, as the NP’s function is presumably related to that of the duplicated gene.

Phylogenomic-guided genome mining can search for divergent enzymes that have possibly evolved from central metabolism to catalyze novel chemical transformations in secondary metabolites. An annotated, functionally characterized enzyme is prerequisite to identifying its divergent clades in a phylogenetic tree [48,51]. However, prior knowledge of a particular enzyme is not necessary in some approaches. Whole BGCs can be phylogenetically analyzed without targeting for specific biosynthesis enzymes, such as by using GCF tools such as BiG-SCAPE [51]. Both resistance-gene-based logic and phylogenomic identification of divergent enzymes can be applied with and without prior knowledge of a particular enzyme.

If the chemical structure of an NP family is known, genome mining can be guided by specific structural features, particularly those in NPs with unprecedented chemistries or potent bioactivities. The presence of specific chemical moieties known to enhance bioactivities, such as the cationic amino acids and thioamides mentioned in this review [54,63], can be directly used for candidate prioritization. Targeting a biosynthesis enzyme with high substrate promiscuity or the ability to catalyze different reactions can be beneficial, as shown in the studies on nisin lanthipeptide heterologous expression [66], cytochrome P450 [51,55], and radical SAMs [57,65].

These studies show that, beyond sequence-similarity-based mining, genes can be targeted according to hypotheses about their functions, evolution, and co-evolution with other genes. The targeted item can also be something other than proteins or peptides, such as the quorum-sensing operons [50,65]. However, a solid hypothesis is not always a prerequisite for starting a large-scale search, because serendipitous discoveries from analyzing a large pool of candidates can be used to generate new hypotheses. An unexpected association of a candidate’s biosynthesis enzymes with known enzymes can prompt its selection, because characterizing it may bring insight into biosynthesis enzyme mechanisms. Selecting putative hits with novel chemical structures or BGC features from global genome mining can expand our understanding of biosynthesis processes and aid in the discovery of novel NP class motifs. Regardless, multiple hypotheses and approaches can be used concurrently to maximize the likelihood of identifying BGCs of interest.

## 8. Future Perspective

There are still many possibilities in the design of NP global genome mining studies in the software and in the overall pipeline. Novel iterative tools to deal with the extensive data from global genome mining have been created, such as BiG-SLiCE [103] and the BiG-FAM database [104], being continuations from the work of BiG-SCAPE, to cluster BGCs into families at faster speeds. The IMG-ABC and MIBiG databases have been updated periodically in recent years, with the former using the newer antiSMASH v5 to replace the previous BGC predictions and the latter applying manual curation to improve their respective schemas and data [6,77]. ClusterFinder has been removed as a default analysis option in antiSMASH v5 onward, and, thus, false positive BGCs resulting from it that are highly unlikely to produce secondary metabolites will not be detected by using the default settings [105]. BGCs detected by ClusterFinder has also been removed from the IMG-ABC database; thus, it should more accurately represent valid predicted BGCs [75,76,77].

Databases specific to different modes of genome mining have also become available. The ARTS-DB contains a repository of putative resistance genes detected by ARTS. ARTS, as a standalone application, can also be used to screen for resistant housekeeping genes in genomes from all bacterial taxa based on gene duplication, horizontal gene transfer (HGT) events, and colocalization with a BGC [106]. To date, ARTS-DB contains putative resistance genes identified through ARTS from more than 70,000 genomes and metagenome-assembled genomes [107]. For phylogenomics-guided genome mining, EvoMining is a tool that allows users to visualize expansion and recruitment events in enzyme families through phylogenetic reconstruction. Users can use their own enzyme, genome, and BGC databases. However, pre-computed databases are also available [71]. The accessibility of different NP genome-mining-specific databases and the availability of the different pipelines to study large-scale BGC data contribute greatly to the performance of global-genome-mining studies. Artificial intelligence (AI) technologies, such as deep neural networks, can offer new perspectives in data mining [108,109].

Besides the genome-mining process, automating other parts of the NP discovery pipeline will make global studies more viable. One common hurdle, again due to the size of the data, is difficulties in detecting the novel compound. The automation of RiPP detection by computationally predicting possible NP structures and fragments has been attempted. This is achieved by generating a library of possible MS spectra of the desired RiPP, from which MS peaks of bacterial culture can be matched to. In a prior-mentioned 2016 study by Skinnider et al. [62], the team conducted gene-guided prediction of their candidate compound’s structure, which allowed them to automate the LC–MS/MS peak searching process by using ion mass predictions based on the predicted structure. Other examples of automated RiPP detection efforts are RiPPquest, pep2path, and pepSAVI-MS [110,111,112]. Automated peptide matching is commonly used in proteomics [113], illustrating how existing methods can be applied to the global genome mining of NPs when being optimized for increased speed, reduced computational load, and high effectiveness.

The age-old bottleneck in the NP discovery pipeline of producing and extracting the NP from bacterial culture has led to the development of various alternative avenues to overcome it. One solution is to synthesize the bioinformatically predicted peptide artificially [114,115]. In RiPP discovery, the promiscuity of tailoring enzymes has been exploited to catalyze chemical transformations on novel precursor peptides, pairing well-expressed tailoring enzymes from characterized BGCs with the novel precursor peptides of interest. This hybrid strategy has been successful in both in vivo and in vitro expression: for in vivo expression, hybrid BGCs containing genes from different BGCs are constructed and expressed [66]; and for in vitro production, the final product is synthesized by in vitro reactions, using purified enzymes [116]. The genetic cloning process for BGCs is another part of the pipeline that can be tackled. After selecting specific BGCs and biosynthesis enzyme genes through global genome mining, the gene can be artificially synthesized. This can bypass the efforts in obtaining and isolating the source microbial strain and enable immediate heterologous expression that simplifies the validation of their biochemical functions. This is particularly useful for sequences derived from metagenome assemblies without strain isolation, as well as for sequences of less-studied bacterial taxa where codon optimization for heterologous expression is being considered.

The characteristics and functions of the source strains that produce bioactive natural products are important information in genome-mining-based discovery of bioactive compounds. For example, thermophilic bacteria may produce more thermostable target peptides or proteins, and, thus, multiple laboratories have prioritized candidate BGCs from thermophilic bacteria, which were achieved through a filtering step post-mining [116,117,118] or only mining from thermophilic bacteria [119]. In addition, candidate BGCs are frequently selected from bacterial strains or organisms that are little-known to avoid NP rediscovery. On the other hand, bacteria that are well-known to be rich natural product producers [86], or bacterial families known to produce a particular type of NP [54], are often the targets for more intensive mining. Well-known taxa which enjoy well-established genetic engineering tools are also favorable for downstream work. One can even combine these two principles in selecting rare and conventional bacteria, as illustrated by Chevrette et al. [120] pursuing *Streptomyces* found on insects, a popular NP powerhouse genus in a little-studied ecological niche. Such biological awareness of the strain can aid in BGC prioritization indirectly.

## Figures and Tables

**Figure 1 marinedrugs-20-00398-f001:**
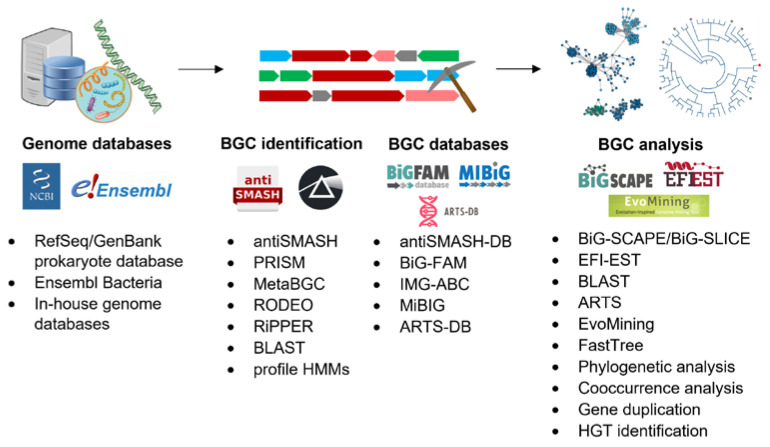
General workflow and examples of bioinformatic tools for natural product discovery guided by large-scale genome mining.

**Figure 2 marinedrugs-20-00398-f002:**
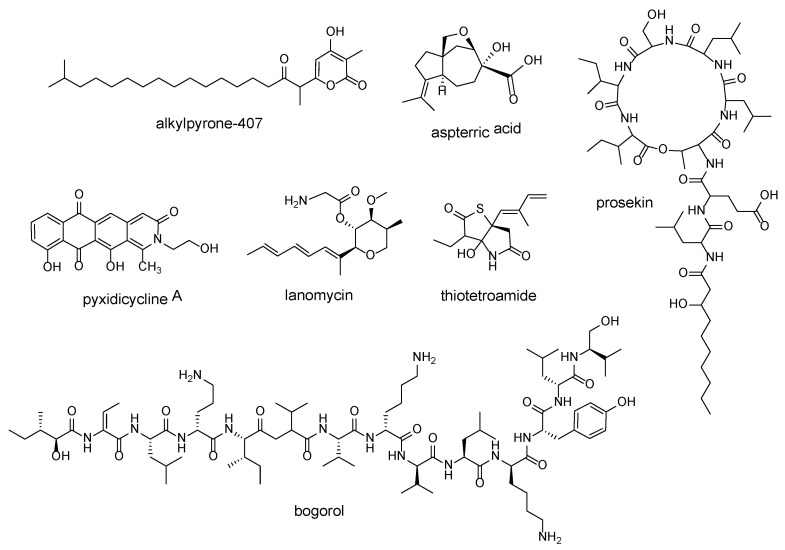
Compounds identified through resistance-gene-guided genome mining and their resistance determinants (in brackets): alkylpyrone-407 and pyxidicycline A (pentapeptide repeat protein (PRP) sequences), aspterric acid (dihydroxyacid dehydratase), prosekin (tripartite efflux system PleABC), lanomycin (lanosterol 14α-demethylase), thiotetroamide (fatty acid synthase), and bogorol (D-stereospecific peptidase).

**Figure 3 marinedrugs-20-00398-f003:**
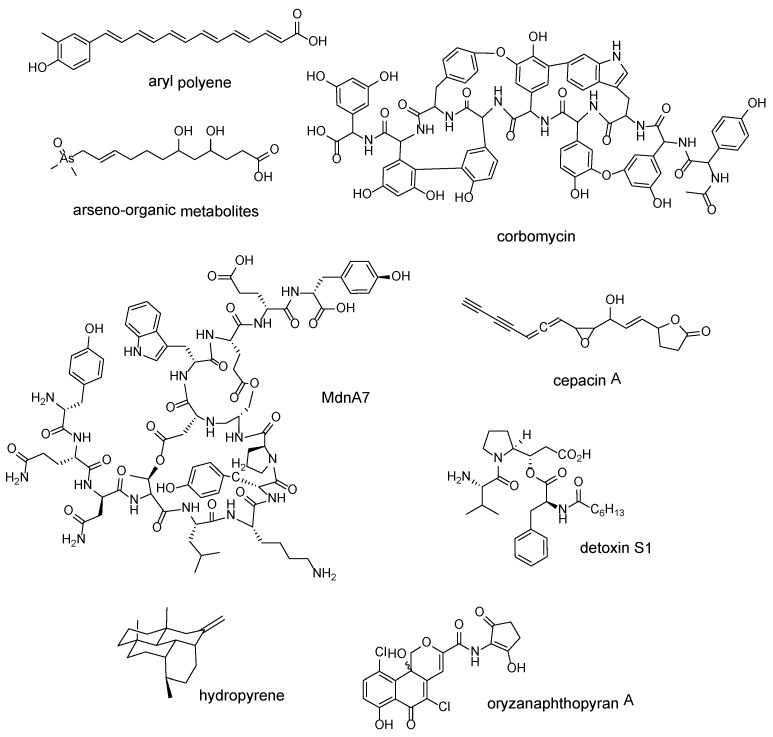
Compounds with novel chemical scaffolds identified through phylogenomics-guided genome mining: aryl polyenes from *Escherichia coli*, arseno-organic metabolites from *Streptomyces lividans*, corbomycin from *Streptomyces* sp. WAC01529, MdnA7 from *Cyanothece* sp. PCC 7822, cepacin A from *Burkholderia* *ambifaria*, detoxin S1 from *Streptomyces* sp. NRRL S-325, hydropyrene from *Streptomyces clavuligerus* ATCC 27064, and oryzanaphthopyran A from *Streptacidiphilus oryzae* CGMCC 4.2012.

**Figure 4 marinedrugs-20-00398-f004:**
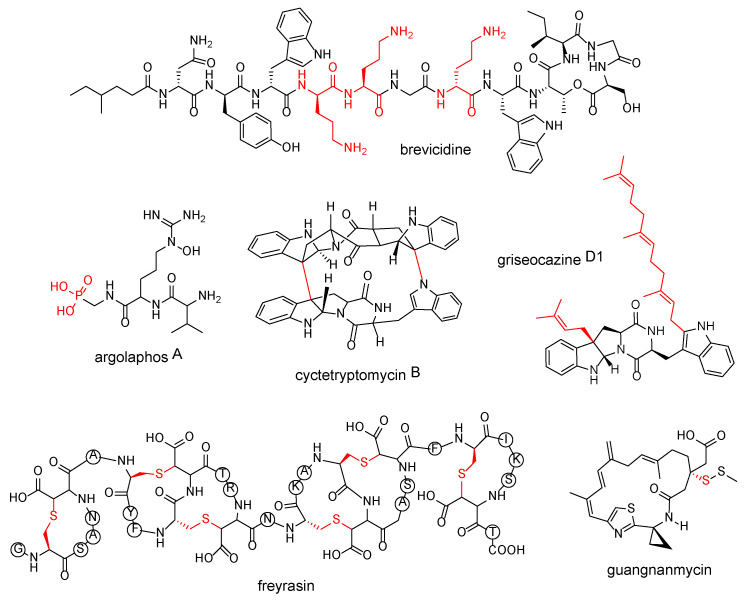
Compounds identified through structure-guided genome mining and specific chemical moieties targeted for (in red): brevicidine (cationic amino acid residues), argolaphos A (phosphonic acid), cyctetryptomycin B (chemical transformations catalyzed by cytochrome P450), griseocazine D1 (prenyl groups), freyrasin (thioether bonds), and guangnanmycin (chemical transformations catalyzed by the DUF–SH didomain).

**Figure 5 marinedrugs-20-00398-f005:**
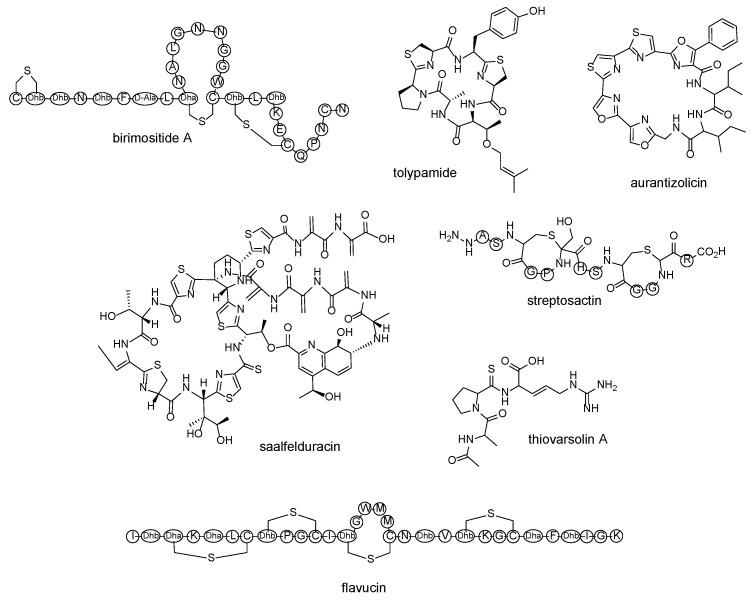
Novel RiPPs identified through large-scale genome mining with unusual biosynthesis and chemical moieities: birimositide *Streptomyces rimosus* subsp. *rimosus* WC3908, tolypamide from *Tolypothrix* sp. PCC 7601, aurantizolicin from *Streptomyces auranticaus* JA 4570, saalfelduracin from *Amycolatopsis*
*saalfeldensis* NRRL B-24474, streptosactin from *Streptococcus thermophilus* JIM 8232, thiovarsolin A from *Streptomyces varsoviensis*, and flavucin from *Corynebacterium lipophiloflavum* DSM 44291.

**Table 1 marinedrugs-20-00398-t001:** Genetic elements and NP features targeted by resistance, phylogenomic, structure, and RiPP-guided genome-mining strategies and the natural products they identified.

Resistance-Gene-Guided			
Resistance Gene(s)	Natural Product	Source Organism	Reference
pentapeptide repeat protein (PRP) sequences	alkylpyrone-407	*Cystobacterineae* strain MCy9487	[39]
	pyxidicycline A	*Pyxidicoccus fallax* An d48	[40]
dihydroxyacid dehydratase	aspterric acid	*Aspergillus terreus* NIH2624	[41]
tripartite efflux system PleABC	prosekin	Pseudomonas prosekii LMG 26867	[42]
lanosterol 14α-demethylase	lanomycin	*Pyrenophora dematioidea* TTI-1096	[43]
fatty acid synthase	thiotetroamide	Streptomyces afghaniensis NRRL 5621	[44]
D-stereospecific peptidase	bogorol	*Brevibacillus laterosporus* DSM 25	[45]
**Phylogenomics-Guided**			
**Sequences Used for Phylogenetic Analysis**	**Natural Product**	**Source Organism**	**Reference**
Whole BGCs of different families	aryl polyenes	*Escherichia coli* CFT073	[46]
“Expanded-then-recruited” enzyme families; 3-carboxyvinyl-phosphoshikimate transferase	arseno-organic metabolites	*Streptomyces lividans* 66	[47]
Each shared gene found in glycopeptide antibiotic-producing BGCs	corbomycin	*Streptomyces* sp. WAC01529	[48]
ATP-grasp ligase	MdnA7	*Cyanothece* sp. PCC 7822	[49]
LuxR	cepacin A	*Burkholderia* *ambifaria* BCC0191	[50]
Whole BGCs containing *tauD* expansion	detoxin S1	*Streptomyces* sp. NRRL S-325	[51]
terpene synthase	hydropyrene	*Streptomyces clavuligerus* ATCC 27064	[52]
chain length factor (CLF) protein	oryzanaphthopyran A	*Streptacidiphilus oryzae* CGMCC 4.2012	[53]
**Structure-Guided**			
**Targeted Chemical Structure**	**Natural Product**	**Source Organism**	**Reference**
cationic amino acid residues	brevicidine	*Brevibacillus laterosporus* DSM 25	[54]
chemical transformations catalyzed by cytochrome P450 on cyclodipeptides	cyctetryptomycin B	*Saccharopolyspora hirsuta* DSM 44795	[55]
prenyl groups on cyclodipeptides	griseocazine D1	*Streptomyces griseocarneus* 132	[56]
thioether bonds	freyrasin	*Paenibacillus polymyxa* ATCC 842	[57]
chemical transformations catalyzed by the DUF–SH didomain	guangnanmycin	*Streptomyces* sp. CB01883	[58]
phosphonic acid	argolaphos A	*Streptomyces monomycini* NRRLB-24309	[59]
**Global Genome Mining for RiPPs**			
Combining the structure-guided strategy with precursor peptide sequence search			
**RiPP Family**	**Natural Product**	**Source Organism**	**Reference**
lanthipeptide	birimositide	*Streptomyces rimosus* subsp. *rimosus* WC3908	[60]
cyanobactin	tolypamide	*Tolypothrix* sp. PCC 7601	[61]
polyoxazole-thiazole-based cyclopeptide	aurantizolicin	*Streptomyces auranticaus* JA 4570	[62]
thiopeptide	saalfelduracin	*Amycolatopsis**saalfeldensis* NRRL B-24474	[63]
thioamitides	thiovarsolin A	*Streptomyces varsoviensis* DSM 40346	[64]
sactipeptide	streptosactin	*Streptococcus thermophilus* JIM 8232	[65]
lanthipeptide	flavucin, agalacticin, etc.	*Corynebacterium lipophiloflavum* DSM 44291	[66]

## Data Availability

Not applicable.

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
