# Peer review of "Targeted Large-Scale Genome Mining and Candidate Prioritization for Natural Product Discovery"

_marinedrugs, 2022, doi:10.3390/md20060398_

Round 1

Reviewer 1 Report

The review entitled “Targeted large-scale genome mining and candidate prioritization for natural product discovery” submitted to “Marine Drugs” by Malit et al, provides information on various approaches for mining BGCs from genomic data. Overall, the manuscript is well written and organized and maybe acceptable for publication.

I just have one minor suggestion for the authors. It will be of great value to the potential readers if the authors could summarize the various approaches and tools described in the paper as a table. For each approach, the tool used and the NP discovered can also be provided as an example in that table.

Author Response

The review entitled “Targeted large-scale genome mining and candidate prioritization for natural product discovery” submitted to “Marine Drugs” by Malit et al, provides information on various approaches for mining BGCs from genomic data. Overall, the manuscript is well written and organized and maybe acceptable for publication.

I just have one minor suggestion for the authors. It will be of great value to the potential readers if the authors could summarize the various approaches and tools described in the paper as a table. For each approach, the tool used and the NP discovered can also be provided as an example in that table.

Thank you very much for your kind comments regarding our manuscript and your suggestion. We also think that the inclusion of a table will greatly benefit the readers, and thus we have summarized all of the strategies stated in this review in Table 1 (Page 4).

We again thank the reviewer for agreeing to review our manuscript.

Reviewer 2 Report

The manuscript by Malit and colleagues reviews approaches to large scale genome mining for natural product biosynthesis operons, together with pertinent examples obtained via different approaches. The manuscript is well written, but I wonder whether it specifically meets the Aims and Scope of “Marine Drugs” as no such examples are highlighted. I would consider it to be a better fit for the MDPI journal “Antibiotics”.

Most of the specific comments below are language or style related.

1. L18 The Abstract changes tense—I would keep the previously used tense and change L18 and 19 to “comment” and “examine”.

2. L28 Do the authors mean “several millenia”---this would be less than 510,000 years and presumably secondary metabolism has been evolving for orders of magnitude longer than that.

3. L81 and throughout Depending on journal style, terms like “in silico”, “in vivo” and “in vitro” should probably be italicized.

4. L112 Please check that NCBI is defined on first use.

5. L133 in previous reviews

6. L135 from a single

7. L151 Although most are enzymes, some resistance proteins bind drugs stoichiometrically (e.g. bleomycin ding proteins in producers)

8. L170, L175, L454 There is various capitalization throughout the manuscript for “Type” –I would change all to capitalized.

9. L172 IC50. The 50 should be subscript

10. L186 The genes for these three proteins are found…

11. L213 Should this be P450 protein (rather than the gene)?

12. L233 Since this is a generalization, perhaps use third person…Genes can be identified….

13. L241 Should “S” be italicized (as in Svedberg)?

14. L243 on a COG..

15 L289 tools could only

16. L352 Streptomyces (in full)

17. L360 a novel…pyran (singular)

18. L375 Strictly speaking, Pythium is not a fungus but a oomycote

19. L444-450. Please include the source organism for this BGC

20.L490 in the genus Bacillus (if the authors are referring to a single genus). If referring to bacilli in general, it should not be italicized

21. L510 RiPP genome mining (to match style used L527)

22. L583 Should shp/rgg be italicized (if referring to genes) or capitalized if referring to proteins?

23. L617 using the heterologous

24. L631 This would not generally be true—for example if the resistance mechanism was drug export

25. L684 Define AI on first use

26. L716 taxa

27. There are various different reference styles…some are abbreviated, some are in full, some have periods after abbreviations, others do not—PNAS is written several different ways, some with USA and some without. Some journal words are capitalized. Others are not.

Reviewer 3 Report

This manuscript reviews aspects of microbial genome mining for natural product discovery focused on methods to identify chemical novelty, biosynthetic pathway novelty, and potent bioactivities. The manuscript is very well written, and the topics covered are currently relevant. This review should be very useful to experts in the field as well as to other scientists interested in learning about the current status of microbial genome mining. I have a few comments to improve the manuscript.

1.       L27-28. “In theory, NPs are structurally optimized …. after several millennia of evolution, …”. It is known that NP BGCs have been evolving over hundreds of millions of years, so “several millenia” does not capture the proper timeframe for BGC evolution. A good reference is: “Waglechner, McArthur & Wright.  Phylogenetic reconciliation reveals the natural history of glycopeptide antibiotic biosynthesis and resistance. Nat Microbiol 2019, 4, 1862-1871”.

2.       L39-40. The conventional abbreviation of “BGC” is for “biosynthetic gene cluster”, not “biosynthesis gene cluster”. I suggest changing it.

3.       L163 and elsewhere in the manuscript. The use of “resistance baits” does not strike me as the best metaphor for the scientific approach. The use of genes that identify where to focus your attention in genome mining might be better described as “molecular beacons” that illuminate genes and gene clusters to focus on.

4.       L286-301. This paragraph on ClusterFinder (55) gives an overly optimistic view of the value of ClusterFinder in identifying novel secondary metabolite BGCs. ClusterFinder was initially an interesting approach to identify truly novel BGCs not discovered over the past 70 years by the pharmaceutical industry. Unfortunately, the limitations and shortcomings of the approach were not articulated well in the original reference (55), and a number of laboratories published papers that grossly overestimated the true numbers of secondary metabolite BGCs in many bacteria. It is now clear that the vast majority of the hypothetical BCGs identified by ClusterFinder do not encode secondary metabolites (Baltz. R.H., Natural product drug discovery in the genomic era: realities, conjectures, misconceptions, and opportunities.  J Ind Microbiol Biotechnol, 2019, 46, 281-299). ClusterFinder has been removed as a common search modality from the most current versions of antiSMASH (5.0 and 6.0) (ref 22) and BGCs identified by ClusterFinder have been removed from IMG-ABC v.5.0 (ref 101) because of the lack of validation of this method to identify legitimate secondary metabolite BGCs (Baltz, R.H., Genome mining for drug discovery: progress at the front end. J Ind Microbiol Biotechnol, 2021, 48, kuab044). These changes in antiSMASH and IMB-ABC have improved the prospects of exploiting microbial genome mining for successful drug discovery. If the authors cite ClusterFinder, they should point out the lack of robust validation of the method for the discovery of novel secondary metabolites, and cite the two Baltz references along with the antiSMASH 6.0 and IMG-ABC v.5.0 references already cited in this manuscript (refs 22 and 101).

5.       L671-674. The comments in 4. should also be used to clarify the statement on L671-674 with more precise information on the removal of Clusterfinder from antiSMASH 5.0 and 6.0, and the total reevaluation of genome sequences with antiSMASH 5.0 (with no ClusterFinder analysis) for the improvement of IMG-ABC v. 5.0 (see the two Baltz references for summaries of the significance of these two changes).

6.       References. Add the references given in comments 1, 4, and 5, along with changes in the text.
